# The magnitude of neonatal asphyxia and its associated factors among newborns in public hospitals of North Gondar Zone, Northwest Ethiopia: A cross-sectional study

**Fitalew Tadele Admasu[1], Biruk Demssie Melese[2], Tadeg Jemere Amare[3], Edget Abebe Zewude[3], Chalachew Yenew Denku[4], Tadesse Asmamaw Dejenie[5]\***

1 Department of Medical Biochemistry, College of Health Sciences, Debre Tabor University, Debre Tabor, Ethiopia, 2 Department of Environmental Health, College of Health Sciences, Debre Tabor University, Debre Tabor, Ethiopia, 3 Department of Medical, Physiology, College of Health Sciences, Debre Tabor University, Debre Tabor, Ethiopia, 4 Department of Social and Public Health (Environmental Health), College of Health Sciences, Debre Tabor University, Debre Tabor, Ethiopia, 5 Department of Biochemistry, School of Medicine, College of Medicine and Health Sciences, University of Gondar, Gondar, Ethiopia

\* as24tadesse@gmail.com

**Data Availability Statement:** All relevant data are within the paper.

## Abstract

### Introduction

Birth asphyxia is a prominent and avoidable cause of infant illness and death worldwide, particularly in underdeveloped countries such as Ethiopia. Early identification and control of the underlying contributory factors would help to alleviate the situation. As a result, the goal of this study was to assess the magnitude and determinants of neonatal asphyxia among live newborns at the northern Gondar public Hospitals in northwest Ethiopia.

### Materials and methods

From April 1 to May 2, 2020, 357 newborns were studied in an institution-based cross-sectional study. The sample size was proportionally distributed among three public hospitals, namely Gondar referral teaching hospital, Debark general hospital, and Kola-Diba District Hospital, which was chosen at random. The number of deliveries given at each hospital six months prior to the data collecting period was used to allocate the hospitals. To get all participants, a systematic random sampling approach was adopted based on hospital delivery registration. The physicians' evaluation of an APGAR score of 7 in the first and fifth minutes of birth was used as the confirmation of birth asphyxia. Data was collected using a standardized and pretested questionnaire. Variables having p-values less than 0.25 were entered into a multivariable logistic regression analysis in the bivariable analysis. At a p-value of 0.05, a statistically significant level was reported.

### Results

As per the study, the total prevalence of neonatal asphyxia was found to be 27.1 (95% CI: 21.4, 32.7). In a multivariable logistic regression analysis, neonates born to rural mothers

**Funding:** The funders had no role in study design, data collection and analysis, decision to publish, or preparation of the manuscript.

**Competing interests:** The authors have declared that no competing interests exist.

**Abbreviations:** AOR, Adjusted Odds Ratio; PROM, premature rupture of membrane; PTB, pulmonary tuberculosis; WHO, World Health Organization.

(AOR = 2.441, 95% CI: 1.137, 5.241), primiparity (AOR = 5.521 95%CI: 1.691, 8.026), premature rupture of membrane, (AOR = 3.202, 95% CI: 1.484, 6.909) and low birth weight (< 2.5kg) (AOR = 3.706, 95%CI: 3.307, 4.152) were all found to be independent predictors of birth asphyxia.

## Conclusion

This study identified that rural residence, primiparity, premature rupture of membrane, and birth weight were found to be the independent predictors of birth asphyxia. The majority of variables that cause birth asphyxia can be controlled.

## Introduction

The first four weeks of a child's life are the most susceptible for survival, with more than one-third of all child fatalities occurring during this period [1]. The neonatal period accounts for 45 percent of all deaths in children under the age of five worldwide [2]. Perinatal asphyxia is responsible for around a quarter of all newborn fatalities globally. Perinatal asphyxia is also responsible for 23% of newborn mortality in low-income countries, with Sub-Saharan Africa accounting for more than two-thirds (38%) of these deaths [2–5]. Birth asphyxia is described by the World Health Organization (WHO) as "the inability to begin and sustain breathing or spontaneous respiration at birth" [6]. It causes the baby to be deprived of oxygen, causing physical injury to essential organs, most commonly the brain [7–9].

The immature newborn brain is highly susceptible to dangerous situations [10]. In most instances, birth asphyxia is not treatable, and those who survive usually suffer from untreatable neurodevelopmental sequelae, like cognitive and motor disabilities, in the short and long term. Around 25% of asphyxia survivors develop hypoxic-ischemic encephalopathy and neurological disorders [11–13]. Perinatal asphyxia has played a significant role in infant morbidity and death across the world [14]. In underdeveloped nations, it is the primary cause of newborn morbidity and mortality, with an estimated incidence of 100-250/1000 live births, contrasted to 5-10/1000 in industrialized countries [15]. Around 4 million babies are born asphyxiated each year, resulting in 2 million neonatal fatalities and intrapartum stillbirths. 23% of all neonatal fatalities and 8% of all deaths in children under the age of 5 are associated with signs of neonatal asphyxia at birth. Almost all newborn mortality occurs in poorer nations, where the majority of babies are born at home with little or no prenatal care and inadequate perinatal care [10].

Various international studies found that multiple risk factors: socio-demographic (mother's age, residence, mother's marital status, educational status, occupation), antepartum (ANC follow up, parity, history of prior neonatal death, pre-eclampsia), intrapartum (prolonged labor, status of amniotic fluid, cephalopelvic disproportion, nature of amniotic fluid), neonatal (preterm babies, birth weight, gestational age are associated with neonatal asphyxia [16, 17]. In underdeveloped nations like Ethiopia, maternity and child health services might avert infant deaths if providing professional care to mothers during pregnancy, as well as during and after birth. To avoid birth asphyxia, minimize mortality, and enhance infant quality of life, it is critical to recognize and manage the factors of birth asphyxia early. However, data on determinants of birth asphyxia are limited in Ethiopia in general, and in the study area in particular and the majority of past research has been conducted at a single institution. Although studies have been conducted in hospitals in the northeastern portion of Amhara, Ethiopia, to the best of

our knowledge, no studies have been conducted in hospitals in the northwestern part of Amhara, Ethiopia. As a result, the purpose of this study was to determine the magnitude and determinants of birth asphyxia in neonates delivered at public hospitals in northwest Ethiopia's north Gondar zone.

## Materials and methods

### Study settings and participants

An institutional-based cross-sectional study was conducted in three hospitals namely (Gondar referral and teaching hospital, Debark general hospital, and Kola Diba District hospital) from April 1 to May 2, 2020. The hospitals are 727, 812, and 757 kilometers from Ethiopia's capital city, Addis Ababa, respectively. These hospitals serve a total of over 2 million people in the catchment regions.

All neonates born with a gestational age of $\geq$ 28 weeks and with their mother were evaluated for eligibility. Newborns with life-threatening abnormalities such as hydrops and cyanotic congenital heart defects were excluded as are already predisposed for asphyxia. Furthermore, preterm babies <28 weeks, those who were seriously ill during the study period, and those neonates who were transferred to advanced care before assessment of APGAR score were also excluded.

A single population proportion formula was used to calculate the sample size. Based on a previous study done in Jimma, Southwest Ethiopia [18], the prevalence of perinatal asphyxia was estimated to be 33%. With a 95% confidence level (the Critical value Zα/2 = 1.96), a 5% margin of error, and adding a 5% non-response rate, the final sample size was (3400.05 + 340) = 357.

$$n = \frac{(Za/2)^2(P)(1-P)}{d^2} = 340$$

Where: n = the required sample size, Z α/2 = the standardized normal distribution curve value for the 95% confidence level, P = the proportion of birth asphyxia among the general population, and d = degree of precision (the margin of error between the sample and population).

Three hospitals were chosen at random from the zone's eight hospitals for this study. The number of study participants was proportionally assigned to each hospital based on the monthly average number of deliveries. In each hospital's delivery unit, a systematic sampling technique with every fifth interval was used to enroll study participants.

### Data collection tools and procedures

Data were collected by using structured questionnaires from both primary and secondary (chart review) sources. A pre-tested structured interviewer based questionnaire was used to collect data on maternal sociodemographic (age, marital status, ethnicity, religion, residence, educational, and occupational status), antepartum (parity, antepartum hemorrhage, co-existing obstetric/medical diseases, and antenatal visits), intrapartum (duration of labor, fetal presentation, mode of delivery, labor attendant, meconium-stained amniotic fluid and premature rupture of membranes), and neonatal related factors (asphyxia, gestational age, birth weight, sex, and birth type) were abstracted using a pre-tested structured checklist from the medical records of pregnant women who gave birth during the data collection period. In each hospital, data was collected by qualified BSc holding midwives and supervised by an MSc holder nurse. The data collection process was evaluated on a daily basis by the primary investigator and

supervisor to ensure data quality and to check for missing information or potential errors. The physicians' evaluation of an APGAR score of 7 in the first and fifth minutes of birth was used as the confirmation of birth asphyxia. The APGAR score was obtained in the birth ward and operation room throughout both day and night.

Data collectors received two days of training on how to gather and record data. The questionnaires were translated into Amharic and then retranslated into English for consistency under the supervision of a pediatrician, and the supervisors and principal investigator checked them daily for accuracy, consistency, and completeness. One week before the actual data collection, the questionnaire was pretested on 17 eligible mother-newborn dyads (5% of the sample size) in Alfa primary hospital.

## Data management and analysis

The data was cleaned up and double-checked before being input into Epi data version 3.1 and exported to SPSS version 20 for analysis. The outcomes of the maternal socio-demographic, antepartum, intrapartum, and neonatal studies were described using descriptive statistics. The degree of relationship between independent and dependent variables was determined using the odds ratio (OR) and confidence interval (CI). Using odds ratios, bivariate logistic regression was used to rank the relative relevance of each exposure variable with the outcome variables. Variables with a P value of less than 0.25 in the bivariate logistic regression analysis were chosen and fitted to the multiple logistic regression analysis to determine the independent effects of each variable. Finally, the Adjusted Odds Ratio with 95% confidence intervals (CI) was used to assess the existence and strength of relationships, with statistical significance reported if $p \leq 0.05$.

## Results

In this study, all 357 mothers agreed to participate, with a 100% response rate. More than half of the respondents, 238(66.67%) were rural inhabitants. The mean age was 26.92 years (SD ± 4.7) of whom 122 (34.17%) belonged to the age category of 20–24 years. Three hundred (84.1%) women were married. In addition, 123 (34.45%) were housewives and only 102 (28.6%) of mothers attended tertiary education. A small number of mothers 23 (6.4%) had a history of adverse pregnancy outcomes (Table 1).

## Antepartum related factors

322(90.2%)] had attended antenatal care at either public hospitals or public health centers. However, only 196(54.9%) moms had four and above antenatal care visits. Of the total respondents, 83(23.2%) of women had Obstetric complications during pregnancy. About 61(17.1%) mothers ever used substances during their gestation (Table 2).

## Intrapartum related characteristics

In our study, 303 (84.87%) of newborns were of vertex presentation, 20 (5.6%) were born by assisted vaginal delivery. The majority 253(70.9%) of the mothers had intrapartum rupture of fetal membranes whereas 104 (29.1%) premature rupture of the membrane was reported among mothers. Finally, 32 (9%) mothers had complicated labor and 103 (28.9%) of births were meconium-stained amniotic fluid following membrane rupture (Table 3).

**Table 1. Socio-demographic factors of mothers who gave live birth at public hospitals of North Gondar zone, 2020 (n = 357).**

| Factors | Response | Frequency | % |
|---|---|---|---|
| Residence | Urban | 119 | 33.33 |
| | Rural | 238 | 66.67 |
| Age (years) | 15–19 | 29 | 8.1 |
| | 20–24 | 122 | 34.17 |
| | 25–29 | 111 | 31.1 |
| | 30–34 | 66 | 18.48 |
| | 35 and above | 29 | 8.1 |
| Marital status | Separated | 25 | 7.0 |
| | Widowed | 32 | 8.9 |
| | Married | 300 | 84.1 |
| Religion | Orthodox | 307 | 85.99 |
| | Muslim | 40 | 11.2 |
| | Others* | 10 | 2.8 |
| Occupation | Housewife | 123 | 34.45 |
| | Self-employee | 86 | 24.1 |
| | Government employee | 99 | 27.7 |
| | Merchant | 45 | 12.6 |
| | Others** | 4 | 1.1 |
| Educational Status | No formal education | 80 | 22.4 |
| | primary | 83 | 23.2 |
| | secondary | 92 | 25.7 |
| | Tertiary and above | 102 | 28.6 |
| Parity | Primiparous | 165 | 46.2 |
| | Multiparous | 192 | 53.8 |
| History of adverse pregnancy outcome | Yes | 23 | 6.4 |
| | No | 334 | 93.6 |

**Table 2. Factors related to the antepartum period among mothers who gave live birth at public hospitals of North Gondar zone, 2020 (n = 357).**

| Factor | Response | Frequency | % |
|---|---|---|---|
| ANC follow up | No, follow up | 35 | 9.8 |
| | 1 time | 33 | 9.2 |
| | 2–3 times | 93 | 26.1 |
| | 4 times and above | 196 | 54.9 |
| Obstetric complication during pregnancy | No, complication | 274 | 76.8 |
| | Preeclampsia/eclampsia | 22 | 6.1 |
| | Antepartum hemorrhage | 14 | 3.9 |
| | Anemia | 38 | 10.6 |
| | Infections | 7 | 1.9 |
| | Gestational diabetes | 2 | 0.6 |
| Any medical illness during pregnancy | Yes | 24 | 6.7 |
| | No | 333 | 93.3 |
| Substance used during pregnancy | Yes | 61 | 17.1 |
| | No | 296 | 82.9 |

**Table 3. Intra-partum related characteristics among mothers who gave live birth at public hospitals of North Gondar zone, 2020 (n = 357).**

| Factor | Response | Frequency | % |
|---|---|---|---|
| Fetal presentation | Vertex | 303 | 84.87 |
| | Malpresentation | 54 | 15.13 |
| Mode of delivery | Spontaneous vaginal delivery | 290 | 81.2 |
| | Cesarean section | 47 | 13.2 |
| | Assisted vaginal delivery | 20 | 5.6 |
| labor Induction | Yes | 14 | 3.9 |
| | no | 343 | 90.1 |
| Labor duration | Normal | 165 | 46.2 |
| | Prolonged | 142 | 39.8 |
| | Precipitated | 50 | 14 |
| Time of membrane rupture | PROM | 104 | 29.1 |
| | Intrapartum | 253 | 70.9 |
| Duration of ROM | Normal | 309 | 86.5 |
| | Prolonged | 48 | 13.5 |
| Status of amniotic fluid at birth | Clear | 247 | 69.2 |
| | Meconium stained | 103 | 28.9 |
| | Bloodstained | 7 | 1.9 |
| Complicated labor | Yes | 32 | 9 |
| | no | 325 | 91 |

## Neonatal related characteristics

189 (53%) of the newborns were males. The mean gestational age at birth was 38.5 (±2.4) weeks and the majority of the newborns 242(67.8%) were term. Moreover, 58(16.3%) of the newborns had low birth weight. At birth, there were 71(19.9%) newborns with health problems. There were 15 (4.2%) twin newborns (Table 4).

**Table 4. Newborn related characteristics among mothers who gave live birth at public hospitals of North Gondar zone, 2020 (n = 357).**

| Factor | Response | frequency | % |
|---|---|---|---|
| Sex | Male | 189 | 53 |
| | Female | 168 | 47 |
| Gestational age at birth | Preterm | 78 | 21.8 |
| | Term | 242 | 67.8 |
| | Post-term | 37 | 1.1 |
| Birth weight | 2.5–4 kg | 276 | 77.3 |
| | <2.5kg | 58 | 16.3 |
| | > 4 kg | 23 | 6.4 |
| Birth outcome | Singleton | 342 | 95.8 |
| | Twin | 15 | 4.2 |
| The neonatal medical problem at birth other than asphyxia | No medical problems | 286 | 80.1 |
| | Neonatal sepsis | 41 | 11.5 |
| | Birth injury | 19 | 5.3 |
| | Congenital malformation | 11 | 3.1 |

## The magnitude of birth asphyxia

The magnitude of birth asphyxia was found to be 97 (27.2%) [95% CI: 21.4%, 32.7%] based on APGAR scoring less than 7 persistently for more than 5 minutes after birth. Most of the asphyxiated neonates had moderate asphyxia 76 (78.4%) whereas 21 (21.2%) neonates had severe birth asphyxia.

## Determinate factors of birth asphyxia

The Bi-variable logistic regression analysis showed that 7 factors namely, residence, gestational age, parity, premature rupture of membrane, birth weight of neonates, antenatal obstetrics complication, mode of delivery were crudely associated with birth asphyxia. However, after statistical adjustments in the final model, gestational age, Antenatal obstetric complications, and mode of delivery were not significant.

Neonates of women from the rural residences had 2.4(AOR = 2.4: 95% CI: 1.4, 5.2) more likelihood of being asphyxiated at birth as compared to the urban residence. Neonates born from primipara mothers were 5.5times (AOR = 5.5: 95%CI: 1.7, 8.1) more prone to be asphyxiated at birth when compared to Multiparous mothers. Furthermore, Neonates born to mothers with premature rupture of fetal membranes were 3.2 times (AOR = 3.2: 95% CI: 1.9, 6.9) more prone to be asphyxiated at birth as compared to those with intrapartum rupture of membranes. Birth asphyxia was 3.8 times (AOR = 3.8, 95%CI: 3.3, 4.2) higher among low-birth-weight neonates than those normal weight neonates (Table 5).

## Discussion

In this study, the prevalence, and predictors of birth asphyxia amongst live births at public hospitals in the south Gondar zone are explored in depth. Birth asphyxia was found to be significantly associated with rural residency, primiparity, preterm ruptures of the membranes, and low birth weight.

In the study, the Perinatal asphyxia was found to be lower (27.1%) than in a study conducted at Dilla Referral Hospital in Southern Ethiopia (32.8%) [19]. However, it was found to be higher than other studies, particularly in India (6.6%), Nigeria (21.1%), Cameroon (8.5%), and also in Ethiopian hospitals of Dire Dawa (12.5%) and Addis Ababa (16.2%) [20, 21]. This discrepancy could be explained by differences in sociocultural characteristics among the study participants. Furthermore, the prevalence of birth asphyxia in our setting was lower than in an Iranian study (58.8%) [22], which could be due to differences in sample size, study setting, and, most importantly, case definition, as our study only used the fifth minute APGAR score less than 7, whereas the Iranian study used more laboratory-based results as diagnostic criteria for birth asphyxia.

Despite the fact that prenatal asphyxia contributes significantly to neonatal morbidity and mortality in underdeveloped countries, evidence is scarce on the factors that predict perinatal asphyxia. There isn't enough evidence to say whether maternal or neonatal variables play a role in the occurrence of birth asphyxia [23, 24].

In an analysis of associated factors, newborns born to women who reside in rural areas had a 2.5 higher risk of having birth asphyxia than those born to women who live in the city. This finding is in line with research was undertaken in Nepal [24], Gondar [25], Dire Dawa [26], and Wollo [27], which found that moms in rural areas had poorer birth outcomes than mothers in urban areas. This could be because the research locations are similar in terms of socioeconomic and lifestyle conditions. The discrepancy between urban and rural residents could be due to a lack of quality pregnancy-related antenatal and prenatal care in rural areas, as well

**Table 5. Bivariable and multivariable logistic regression analysis of factors associated with birth asphyxia among live births at public hospitals of North Gondar zone, 2020 (n = 357).**

| Factor | Birth asphyxia | | 95% CI | | P-value |
|---|---|---|---|---|---|
| | Asphyxiated (n = 97) | Not asphyxiated (n = 260) | COR (95%CI) | AOR (95%CI) | |
| Residence | | | | | |
| Urban | 38 | 200 | 1.0 | 1.0 | |
| Rural | 59 | 60 | 1.8(0.9, 3.4) | 2.4(1.4, 5.2) | 0.036 |
| parity | | | | | |
| Primiparous | 57 | 108 | 4.7(1.7, 12.8) | 5.5(1.7, 8.1) | 0.001 |
| Multiparous | 40 | 152 | 1.0 | 1.0 | |
| Birth weight | | | | | |
| < 2.5 kg | 45 | 13 | 13.5(1.5,23.7) | 3.8(3.3, 4.2) | 0.011 |
| 2.5-4kg | 39 | 237 | 1.0 | 1.0 | |
| > 4 kg | 13 | 10 | 5.3(0.7, 42.3) | 2.7(1.1,5.9) | |
| Membrane rupture | | | | | |
| PROM | 63 | 41 | 2.4(1.2, 4.5) | 3.2(1.9, 6.9) | 0.031 |
| Intrapartum | 34 | 219 | 1.0 | 1.0 | |
| Gestational age | | | | | |
| Preterm | 36 | 42 | 3.8(0.4, 33.1) | 8.7(0.6, 13.76) | 0.098 |
| Term | 47 | 195 | 1.0 | 1.0 | |
| Post-term | 14 | 23 | 1.8(0.2, 16.6) | 3.6(0.2, 52.9) | 0.779 |
| Antenatal obstetric complications | | | | | |
| Yes | 65 | 18 | 16.8(10.5,26.8) | 6.5 (0.6, 1.6) | 0.08 |
| No | 32 | 242 | 1.0 | 1.0 | |
| Mode of delivery | | | | | |
| SVD | 45 | 245 | 1.0 | 1.0 | |
| Assisted vaginal | 13 | 7 | 3.9 (2.4, 6.3) | 6.2 (2.4, 14.6) | 0.517 |
| CS | 39 | 8 | 9.5 (5.5, 15.1) | 6.7 (0.7, 2.0) | 0.065 |

CS = Cesarean section.

as the availability and distance of health facilities, a lack of information about antenatal and prenatal care, transportation issues, and a high load of housework.

In comparison to multiparous women, neonates delivered to primiparous mothers were shown to be five times more likely to be asphyxiated at delivery. The finding is consistent with research conducted in Pakistan, Nigeria, and central Tigray, which found that being primipara is one of the independent factors of newborn asphyxia [28, 29]. The consistency can be justified by the fact that primipara mothers are more likely to be found at a younger age and may be more prone to malpresentation, prolonged obstructed labor, and the mother who is experiencing labor and delivery for the first time may take longer to negotiate with the pelvic brim and may fail to progress in labor, resulting in the delivery of an asphyxiated neonate.

In this study, low birth weight was also founded to be a major factor of birth asphyxia. Low—birth—weight neonates were 3.72 times more likely than normal birth weight neonates to suffer from birth asphyxia. These findings matched those of research undertaken at Gondar Referral Hospital [12], Jimma hospital [18], and Iran [30]. This could be explained by the fact that a high proportion of small babies are born prematurely and lack sufficient surfactant, resulting in breathing difficulties, and low-birth-weight neonates typically have pulmonary immaturity and weak respiratory muscle strength.

Furthermore, compared to those delivered to mothers with intrapartum rupture, neonates born to mothers with premature rupture of fetal membranes (PROM) were 3.2 times more likely to be asphyxiated at birth. This finding is in line with studies in Cameroon [31], Uganda [32], and Al-Diwaniya Teaching Hospital in Saudi Arabia [33]. A prospective case-control study on term newborns in Yaoundé, Cameroon, found a link between premature ruptured membranes and delivery hypoxia [34].

## Conclusions

In conclusion, the prevalence of birth asphyxia is comparable to previous studies in underdeveloped nations. Rural inhabitants, primiparity, preterm rupture of membranes, and low birth weight were all found to be independent predictors of birth asphyxia. As a result, health care personnel, particularly those in labor and delivery wards, must pay more attention to complex labors to foresee and prevent birth asphyxia.

## Acknowledgments

The authors would like to thank all respondents for their willingness to participate in the study, the data collectors, and the supervisors. We are also very glad to forward our special thanks to the employee in the selected regional hospitals and health bureau office for their cooperation.

## Author Contributions

**Conceptualization:** Fitalew Tadele Admasu.

**Data curation:** Fitalew Tadele Admasu.

**Formal analysis:** Fitalew Tadele Admasu, Tadeg Jemere Amare.

**Investigation:** Biruk Demssie Melese, Tadeg Jemere Amare.

**Methodology:** Fitalew Tadele Admasu, Tadeg Jemere Amare, Edget Abebe Zewude, Tadesse Asmamaw Dejenie.

**Resources:** Chalachew Yenew Denku.

**Supervision:** Biruk Demssie Melese, Chalachew Yenew Denku.

**Visualization:** Tadesse Asmamaw Dejenie.

**Writing – original draft:** Fitalew Tadele Admasu, Chalachew Yenew Denku.

**Writing – review & editing:** Edget Abebe Zewude, Tadesse Asmamaw Dejenie.

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
