## [Decision Letter · Decision Letter 0]

13 Jan 2022

PONE-D-21-24852The magnitude of Neonatal Asphyxia and its Associated Factors among newborns; A cross-sectional studyPLOS ONE

Dear Dr. Dejenie 

Thank you for submitting your manuscript to PLOS ONE. After careful consideration, we feel that it has merit but does not fully meet PLOS ONE’s publication criteria as it currently stands. Therefore, we invite you to submit a revised version of the manuscript that addresses the points raised during the review process.

We look forward to receiving your revised manuscript.

Kind regards,

Francesca Crovetto

Academic Editor

PLOS ONE

https://journals.plos.org/plosone/s/file?id=ba62/PLOSOne_formatting_sample_title_authors_affiliations.pdf”

2. Thank you for submitting the above manuscript to PLOS ONE. During our internal evaluation of the manuscript, we found significant text overlap between your submission and the following previously published works, some of which you are an author.

https://journals.plos.org/plosone/article?id=10.1371%2Fjournal.pone.0226891

https://bmcpregnancychildbirth.biomedcentral.com/articles/10.1186/s12884-020-03348-2

Please revise the manuscript to rephrase the duplicated text, cite your sources, and provide details as to how the current manuscript advances on previous work. Please note that further consideration is dependent on the submission of a manuscript that addresses these concerns about the overlap in text with published work.

**Comments to the Author**

**Reviewer #1: **I read with great interest the manuscript, which falls within the aim of this Journal. In my honest opinion, the topic is interesting enough to attract the readers’ attention. Nevertheless, authors should clarify some points and improve the discussion, as suggested below.

Authors should consider the following recommendations:

- Manuscript should be further revised in order to correct some typos and improve style.

- I would recommend to highlight clearly other conditions that may cause adverse obstetric outcomes, including neonatal asphyxia, such as advanced maternal age (PMID: 25027820), pre-eclampsia (PMID: 26512423; PMID: 32283429) and obstructed labor (PMID: 31823037; PMID: 32307556 ).

**Reviewer #2: **This is a cross sectional study analyzing cases of fetal asphyxia in northwest Ethiopia.

The article has some inconsistencies. The most evident are:

- In the abstract you say " The diagnosis of birth

asphyxia was confirmed based on the physician’s diagnosis of an APGAR score < 7 in

the 1 st and 5 th minutes of birth". IN the text you say "The diagnosis of birth asphyxia was made based on the fifth-minute APGAR scores of < 7 ". Please correct

- Regarding the antepartal visits, you reported in the text that 59.1% of patients had 4 or more visits, while in the table they are 54.09%

- Married women are 325 (91.1%) in the text and 32 (8.9%) in the table

I recommend reviewing all correspondences between text and tables.

In table 4 you wrote about "Neonatal medical problem at birth" that 286 babies had no problems, 41 sepsis, 19 birth injury and 11 congenital malformation. It is not clear where the 97 cases of childbirth asphyxia fit in.

In lines 192-193 In lines 192-193 you divide fetal asphyxia into moderate and severe but it is not defined by what criteria this subdivision is carried out.

---

## [Author Response · Author response to Decision Letter 0]

9 Feb 2022

January 28, 2022

Rebuttal letter

PONE-D-21-24852

The magnitude of neonatal asphyxia and its associated factors among newborns; a cross-sectional study

Dear Editors and Reviewers,

We are grateful for the critical review and constructive suggestions to improve our manuscript. Based on the comments and suggestions, we have made corrections and modifications and provided point-by-point responses to comments and suggestions. Please, find our responses in a green mark to comments/suggestions presented by academic editors and reviewers marked in yellow. Below are our responses to each point raised by the academic editor and reviewers.

Best regards!

Tadesse Asmamaw Dejenie, on behave of all authors

Response to academic editor’s comment

Comment 1: Please ensure that your manuscript meets PLOS ONE's style requirements, including those for file naming

Author response: Thank you so much for sending the link to the PLOS ONE style templates; we have taken your advice and updated our manuscript to meet the style requirements of PLOS ONE. Please double-check it in the revised manuscript.

Comment 2: Thank you for submitting the above manuscript to PLOS ONE. During our internal evaluation of the manuscript, we found significant text overlap between your submission and the following previously published works, some of which you are an author. 

Author response: We'd like to thank you for your input. We've taken your advice and thoroughly reread our amended manuscript. Please see the revised manuscript for further information.

Comment 3: We note that you have stated that you will provide repository information for your data at acceptance. Should your manuscript be accepted for publication, we will hold it until you provide the relevant accession numbers or DOIs necessary to access your data. If you wish to make changes to your Data Availability statement, please describe these changes in your cover letter and we will update your Data Availability statement to reflect the information you provide.

Author response: We'd want to thank you one more for alerting us. Please correct, as the datasets used in this work are available upon reasonable request from the corresponding author.

Response to Reviewers’ comment

Reviewer #1: I read with great interest the manuscript, which falls within the aim of this Journal. In my honest opinion, the topic is interesting enough to attract the readers’ attention. Nevertheless, authors should clarify some points and improve the discussion, as suggested below.

Authors should consider the following recommendations:

- Manuscript should be further revised in order to correct some typos and improve style.

- I would recommend highlighting clearly other conditions that may cause adverse obstetric outcomes, including neonatal asphyxia, such as advanced maternal age (PMID: 25027820), pre-eclampsia (PMID: 26512423; PMID: 32283429) and obstructed labor (PMID: 31823037; PMID: 32307556 ).

Author response: We are really grateful for your suggestions. We have acknowledged and adjusted your helpful suggestions. Tables 3 and 4 of the revised manuscript address some of your issues. For further information, please kindly check the revised manuscript.

Reviewer #2: 

Comment 1: The article has some inconsistencies. The most evident are:

- In the abstract you say " The diagnosis of birth asphyxia was confirmed based on the physician’s diagnosis of an APGAR score < 7 in the 1st and 5th minutes of birth". IN the text you say "The diagnosis of birth asphyxia was made based on the fifth-minute APGAR scores of < 7 ". Please correct

Author response: We'd like to express our gratitude for your insightful remarks and suggestions. We've taken note of the suggestions and made the necessary revisions.

Comment 2: Regarding the antepartal visits, you reported in the text that 59.1% of patients had 4 or more visits, while in the table they are 54.09%

Author response: We appreciate your thoughtful and helpful remarks once again. We are perplexed as to why this issue has arisen. This is most likely due to a technical glitch. In any case, we took your suggestions and made the necessary changes.

Comment 3: Married women are 325 (91.1%) in the text and 32 (8.9%) in the table

I recommend reviewing all correspondences between text and tables.

In table 4 you wrote about "Neonatal medical problem at birth" that 286 babies had no problems, 41 sepsis, 19 birth injury and 11 congenital malformation. It is not clear where the 97 cases of childbirth asphyxia fit in.

Author response: We value your critical opinion. In fact, when we assessed the neonatal medical condition at birth, we did not consider neonatal asphyxia; instead, we assessed medical problems other than asphyxia.

Comment 4: In lines 192-193 you divide fetal asphyxia into moderate and severe but it is not defined by what criteria this subdivision is carried out.

Author response: Fetal asphyxia was classified using the Standard Treatment Protocol for Management of Common Newborn Conditions in Small Hospitals, which is based on WHO Guidelines.

---

## [Editor Report · Decision Letter 1]

18 Feb 2022

The magnitude of neonatal asphyxia and its associated factors among newborns in public hospitals of North Gondar Zone, Northwest Ethiopia: A cross-sectional study

PONE-D-21-24852R1

Dear Dr. Tadesse Asmamaw, 

We’re pleased to inform you that your manuscript has been judged scientifically suitable for publication and will be formally accepted for publication once it meets all outstanding technical requirements.

Kind regards,

Francesca Crovetto

Academic Editor

PLOS ONE

---

## [Editor Report · Acceptance letter]

22 Feb 2022

PONE-D-21-24852R1 

The magnitude of neonatal asphyxia and its associated factors among newborns in public hospitals of North Gondar Zone, Northwest Ethiopia: A cross-sectional study 

Dear Dr. Dejenie:

I'm pleased to inform you that your manuscript has been deemed suitable for publication in PLOS ONE. Congratulations! Your manuscript is now with our production department. 

Kind regards, 

on behalf of

Dr. Francesca Crovetto 

Academic Editor

PLOS ONE